# Exploration of the Misfolding Mechanism of Transthyretin Monomer: Insights from Hybrid-Resolution Simulations and Markov State Model Analysis

**DOI:** 10.3390/biom9120889

**Published:** 2019-12-17

**Authors:** Shuangyan Zhou, Jie Cheng, Ting Yang, Mingyue Ma, Wenying Zhang, Shuai Yuan, Glenn V. Lo, Yusheng Dou

**Affiliations:** 1Chongqing Key Laboratory on Big Data for Bio Intelligence, Chongqing University of Posts and Telecommunications, Chongqing 400065, China; zhousy@cqupt.edu.cn (S.Z.); 15569603146@163.com (J.C.); mamingyue@cqupt.edu.cn (M.M.); zhangwenying@cqupt.edu.cn (W.Z.); 2Department of Chemistry and Physical Sciences, Nicholls State University, P.O. Box 2022, Thibodaux, LA 70310, USA; glenn.lo@nicholls.edu (G.V.L.); yusheng.dou@nicholls.edu (Y.D.)

**Keywords:** protein misfolding, transthyretin amyloidosis, molecular dynamics simulation, Markov state model

## Abstract

Misfolding and aggregation of transthyretin (TTR) is widely known to be responsible for a progressive systemic disorder called amyloid transthyretin (ATTR) amyloidosis. Studies suggest that TTR aggregation is initiated by a rate-limiting dissociation of the homo-tetramer into its monomers, which can rapidly misfold and self-assemble into amyloid fibril. Thus, exploring conformational change involved in TTR monomer misfolding is of vital importance for understanding the pathogenesis of ATTR amyloidosis. In this work, microsecond timescale hybrid-resolution molecular dynamics (MD) simulations combined with Markov state model (MSM) analysis were performed to investigate the misfolding mechanism of the TTR monomer. The results indicate that a macrostate with partially unfolded conformations may serve as the misfolded state of the TTR monomer. This misfolded state was extremely stable with a very large equilibrium probability of about 85.28%. With secondary structure analysis, we found the DAGH sheet in this state to be significantly destroyed. The CBEF sheet was relatively stable and sheet structure was maintained. However, the F-strand in this sheet was likely to move away from E-strand and reform a new β-sheet with the H-strand. This observation is consistent with experimental finding that F and H strands in the outer edge drive the misfolding of TTR. Finally, transition pathways from a near native state to this misfolded macrostate showed that the conformational transition can occur either through a native-like β-sheet intermediates or through partially unfolded intermediates, while the later appears to be the main pathway. As a whole, we identified a potential misfolded state of the TTR monomer and elucidated the misfolding pathway for its conformational transition. This work can provide a valuable theoretical basis for understanding of TTR aggregation and the pathogenesis of ATTR amyloidosis at the atomic level.

## 1. Introduction

Transthyretin (TTR) is a globular protein whose misfolding and amyloid aggregation is related to amyloid transthyretin (ATTR) amyloidosis that consists of wild-type ATTR (ATTRwt) amyloidosis and variant ATTR (ATTRv) amyloidosis [1]. ATTRwt amyloidosis, traditionally called senile systemic amyloidosis, is caused by misfolding and aggregation of wild-type TTR [2], especially in people older than 65, while ATTRv amyloidosis, also known as familial amyloidotic polyneuropathy, and familial amyloidotic cardiomyopathy are due to various mutations [3,4] which can accelerate the misfolding and aggregation of TTR. A common pathological feature of different ATTR amyloidosis is the deposition of TTR amyloid fibril in various tissues, which causes significant damage to related organs [5,6]. However, little is known about the pathogenesis of ATTR amyloidosis, which largelycontributes to the lack of clarity about the misfolding and aggregation mechanism of TTR.

TTR is a 55 kDa homo-tetrameric protein, which acts as a carrier of the thyroid hormone thyroxine and retinol-binding protein in vivo. The dissociation of the tetramer into TTR monomers is thought to be the rate-limiting step for fibril formation [7,8,9]. Once the TTR tetramer is dissociated into monomers, misfolding rapidly occurs and leads to aggregation into amyloid fibrils. The native state TTR monomer is a 127 amino acid protein with eight β-strand chains and a short α-helix motif. As shown in Figure 1, these eight β-strand chains form two four-β-stranded anti-parallel sheets, which are called DAGH and CBEF, respectively. Despite the β-sheet-rich feature of the TTR monomer, it is the only form of TTR that misfolds. Therefore, it is critical to explore the misfolding mechanism of the TTR monomer, as it may provide significant clues for understanding the pathogenesis of ATTR amyloidosis.

In recent decades, the TTR monomer misfolding mechanism has been extensively investigated by experimental methods and molecular dynamics (MD) simulations. Experimental studies by Quintas et al. suggest that the TTR monomer tends to form partially unfolded intermediates which precede the protofibril formation [10]. Lim et al., using nuclear magnetic resonance (NMR) relaxation dispersion, found that localized structural fluctuations of TTR, especially in the DAGH sheets, can promote conformational change to form amyloid fibril [11,12]. Their work also suggests that the amyloidogenic precursor states of wild-type TTR may adopt native-like β-sheet conformations [13]. Dasari et al. recently reported that depending on conditions, the aggregation pathway may involve native-like β-sheet intermediates or largely unfolded states [14]. Interestingly, recent studies reported by Koike et al. found that the morphology of TTR amyloid fibrils between early- and late-onset ATTR amyloidosis patients shows a significant difference [15,16], which implies that the TTR at different stages may have different misfolding and aggregation mechanisms. These experimental studies have provided deep insights into the overall structural characteristics of the TTR monomer misfolding mechanism. However, atomic-level information is still hard to obtain due to the limited spatiotemporal resolution of experimental techniques.

MD simulation has been a valuable technique for obtaining detailed information about structural changes in biomolecules to complement experimental observations. MD simulation has also been widely applied to investigate structural dynamics of TTR in recent years [17,18]. For example, based on information from neutron crystallography, native mass spectrometry, and MD simulation, Yee et al. proposed that TTR can form amyloid fibrils via a parallel equilibrium of partially unfolded species, and that the unfolding process is initiated in a CD loop [19]. Rodrigues also reported a premature disruption, with displacement of strands D and C from the core of the TTR monomer, during the unfolding process [20]. In addition, Armen, et al. simulated the structural dynamics of TTR monomer at different temperatures and acidic pH. They found the CBEF sheet was disrupted while an α-sheet structure was formed in DAGH at low pH, which they proposed to be a key pathological conformation during TTR amyloidogenesis [21].

While significant progress has been made in the computational investigation of the TTR misfolding mechanism, the misfolding mechanism of TTR monomer remains unclear and many questions regarding conformational transitions during TTR unfolding still need to be answered. These include: (1) What happens at the atomic level? (2) What kind of pathway is preferred? (3) What are the structural features of intermediates? Moreover, the protein folding and unfolding processes generally occur in the microsecond to millisecond time scales [22,23] whereas simulation times in the aforementioned MD studies are limited to only tens to hundreds of nanoseconds, a time scale too short to reveal biologically relevant structural changes for systems, such as TTR, that have more than one hundred residues. 

In this work, to investigate the mechanism and identify potential pathways and intermediates of the TTR monomer misfolding, microsecond time-scale simulations were performed. To reduce computational costs, a hybrid-resolution model, the protein in atomistic details coupled with coarse-grained environment (PACE) was employed. PACE is a hybrid united-atom and coarse-grained force field developed for protein folding [24,25,26], for which the protein is represented in atomic detail (united-atom model) while the solvent is described by a Martini coarse-grained solvent model. The PACE model has been previously shown to reliably maintain the native structure of proteins that are either α-helix-rich or all-β [27]. This model has also been widely used to study protein folding [28,29]. Meanwhile, the Markov state model (MSM) is applied to analyze the mechanisms of TTR conformational transitions and to identify conformational states during TTR monomer misfolding. MSM is a method which can predict protein dynamics over a long time scale from many short discrete simulations [30,31]. More importantly, the MSM provides a way to model kinetic networks between different states in conformational space based on kinetic criteria [32], from which we can gain insight regarding the conformational transition pathway. By combining microsecond hybrid-resolution simulation and MSM analysis, we sought to unravel the potential misfolding mechanism of the TTR monomer, as it may provide a valuable theoretical basis for understanding the pathogenesis of TTR-related diseases. 

## 2. Materials and Methods

### 2.1. Preparation of Initial Structures

In this work, the initial TTR monomer structure was extracted from the X-ray crystal structures of human TTR tetramer taken from RSCB Protein Data Bank (PDB ID: 4PVM) [33]. As shown in Figure 1, the TTR monomer contains eight β-sheet strands (labeled A through H), which constitute the DAGH and CBEF sheets. The input files for subsequent hybrid-resolution simulation were generated from the CHARMM-GUI [34] website using the PACE CG Solution Builder input generator (http://www.charmm-gui.org/?doc=input/pacecg.solution) [27]. To validate the reliability of applying the PACE model on TTR monomer simulation, an all-atom force field, with CHARMM 36 force field [35], was also simulated for comparison. The input files for the all-atom simulation were prepared using visual molecular dynamics (VMD) [36] (version 1.9.3).

### 2.2. Molecular Dynamics Simulations

All simulations, including hybrid-resolution and all-atom simulation, were performed in NAMD 2.9 [37] with a minor modification for using PACE parameters. For hybrid-resolution simulations, the TTR monomer was represented by the PACE model which retains heavy atoms and most of the polar hydrogen atoms [24]. The monomer was solvated in a box with Martini water and five Na^+^ were added to maintain electroneutrality. Switch functions were employed to Coulomb potentials from 0 to 12 Å, and Lennard-Jones potentials from 9 to 12Å. During the simulation, pressure was maintained at 1 atm by a Nosé–Hoover Langevin piston barostat with period of 200 fs and a decay rate of 100 fs. The Langevin thermostat method was used to maintain a constant temperature of 300 K. The simulation time step was set to 5 fs, which is a typical value used in PACE simulations [24]. Three parallel runs were performed to confirm the reliability; each run was 2 μs long. For the all-atom simulation, the monomer was solvated in a cubic box using the TIP3P water model [38]. A cut-off of 12 Å was used for Lennard-Jones interaction and the particle mesh Ewald (PME) method [39] was used to treat electrostatics. The simulation was carried out with 100 ps minimization followed by 1 ns structure equilibration. Finally, a 500 ns MD simulation with a 2 fs time step was run. All simulations were performed at the physiological condition with pH of 7.

### 2.3. Markov State Model and Construction and Validation

A general problem for applying MD simulation to explore the transition mechanism of protein folding is the challenge to reach a biologically relevant time scale. The MSM has been shown to be an efficient solution to solve such rare-event simulation problems. It can be used to extract equilibrium and dynamic information from general MD simulations, and it also enables the prediction of long time-scale kinetics from many short simulation trajectories [40,41]. With MSM, the conformational space is divided into a number of metastable states that share similar structural and kinetic properties. It is also possible for the MSM to provide a way to model kinetic networks between different metastable states to uncover potential transition pathways. Typically, a MSM is a memoryless process which depends only upon the present state. These dynamics can be modeled by a first-order master equation to give global long time scale dynamics:(1)$$P(n\tau )={\left[T(\tau )\right]}^{n}P(0)$$
where P(nτ) is the vector of state populations at time nτ and T(τ) is the transition probability matrix with lag time of τ. Implied time scales $t_{i}(\tau )$ can be calculated to determine the appropriate lag time and to check if the model is Markovian as well. The calculation formula of $t_{i}(\tau )$ is as follows: (2)$$t_{i}(\tau )=-\frac{\tau }{\ln\lambda_{i}(\tau )}$$
where τ is the lag time and $\lambda_{i}(\tau )$ is the *i*th largest eigenvalue of the transition matrix with lag time τ. If all the microstates generated are ideally Markovian, the implied time scales should remain constant regardless of the choice of lag time [42].

To construct the MSM, we first clustered all sampled conformations into microstates by using the *k*-means clustering method [43] and employing principal components to measure structural similarity between conformations [44]. Before clustering, the Cα distances of backbone atoms were selected as input features to reduce the dimension of conformational space. To remove the influence of the rotation and translation of the protein, all conformations were aligned to the first frame. After clustering, the implied time scales were calculated to determine lag time and number of macrostates for MSM construction. Based on the plot of implied time scales, we then lumped all related microstates into macrostates at a certain selected lag time based on the kinetic similarity using the Perron Cluster Analysis method (PCCA+) [45]. To ensure the constructed MSM is indeed Markovian, the Chapman–Kolmogorov (C–K) test was further calculated since the convergence of lag time is not sufficient for validation of Markovianity. The C–K test is calculated according to the following equation:(3)$$T(n\tau )\approx T{\left(\tau \right)}^{n}$$
where T(τ) is the transition probability matrix with selected lag time τ and *n* is an integer number of steps. The C–K compares the probability of the protein staying in a certain state predicted from the constructed MSM, with that of MD trajectories at increasing time steps. All construction and validation of the MSM was performed in the PyEMMA software package [46]. 

### 2.4. Dynamical Cross-Correlation Map Analysis

Dynamic cross-correlation (DCC) is a method used to describe the fluctuations in cross correlations and the domain motions of Cα atoms during simulation [47]. This method has been largely applied to quantify the correlation coefficients of motions between atoms [48,49]. The correlation coefficient $C_{ij}$ of the cross correlation matrix for two Cα atoms *i* and *j* is defined by the following equation [50]:(4)$$C_{ij}=\frac{\langle \Delta r_{i}\Delta r_{j}\rangle }{{\left[\langle \Delta r_{i}\Delta r_{i}\rangle \langle \Delta r_{j}\Delta r_{j}\rangle \right]}^{1/2}}$$
where $\Delta r_{i}$ and $\Delta r_{j}$ represent the displacement vectors of atoms *i* and *j* with respect to their mean position, and the <…> denotes trajectory averages. Highly correlated motions are denoted by positive $C_{ij}$, indicating residues move in the same direction, while negative $C_{ij}$ denote anti-correlated motions with residues moving in the opposite direction.

## 3. Results and Discussion 

### 3.1. Comparison of PACE Simulation and All-Atom Simulation

Three parallel runs were performed for the hybrid-resolution PACE simulation; the results (Run1, Run2, and Run3) are summarized in Figure 2 and Figure 3. Each PACE run was 2 μs long. For comparison, a 500 ns all-atom simulation (Run-AA) was also performed to check if the PACE model was accurate. To monitor the structural stability of simulated systems, we calculated the root-mean-square deviation (RMSD) of Cα atoms of the TTR monomer from residues 10 to 124; results are shown in Figure 2a. The top panel inset in Figure 2a shows larger RMSDs for PACE runs compared to Run-AA (~1.6 Å). However, the RMSDs for the PACE runs are fairly stable at ~3 Å during the first 500 ns. This means that the native structure of the TTR monomer is maintained during the first 500 ns. 

Consistently, further root-mean-square fluctuations (RMSFs) analysis of Cα atoms of conformations from 400 to 500 ns showed similar fluctuation for all runs, suggesting the similar structural flexibility for PACE and all-atom simulations. As shown in Figure 2b, the flexibility regions corresponded to loops in the TTR monomer, viz., residues 20-28 (A-B loop), residues 37–40 (B-C loop), residues 55–67 (strand D and D-E loop), residues 79–90 (E-F loop), residues 98–102 (F-G loop), and residues 113–117 (G-H loop). Interestingly, short β-strand D from residues 55 to 56 also show quite large fluctuation during simulations, indicating an instability for this β-strand. This strand has been experimentally shown to have native instability [13]. Hence, we can conclude that the hybrid-resolution PACE simulation was reliable in our work.

Notably, it is worth noting that there was an obvious increase in RMSDs after ~1 μs as shown in Figure 2a for all PACE runs, which indicated the potential structural transition of TTR monomer. Therefore, we then performed a secondary structural analysis to monitor structural change during simulations. As shown in Figure 3, the helix structure (residue 75–83) was well maintained in all runs. For the eight β-strands, the most obvious structural change occurred in strand D in both PACE runs and in Run-AA; in each case, the strand was partially or totally unfolded into disordered coil and turn structures. These results are consistent with the above RMSF analysis showing large structural flexibility of strand D. It is clear from Figure 3 that disruption in the DAGH sheet was more significant compared to the CBEF sheet, suggesting the larger structural dynamics of DAGH sheet. Importantly, the instability of the DAGH sheet has been previously shown experimentally [11,51]. For the all-atom run, the structural dynamics of the DAGH sheet were not as obvious as in the PACE runs. This may have been due to the shorter simulation time. On the other hand, a different force field and solvent model may influence the structural dynamics of proteins more or less, e.g., the CHARMM 22 force field overestimates the stability of helical structure [52], while the AMBER force field FF96 shows bias favoring extended β-structure [53].

Overall, the aforementioned analyses suggest that PACE simulations are reliable and do capture obvious structural changes in the TTR monomer. But this does not provide adequate information about the pathway for TTR misfolding as well as the key intermediates during the misfolding process. To solve this problem, a MSM was further constructed based on the discrete trajectories of three parallel PACE runs over a 6 μs timescale. 

### 3.2. Validation of Constructed Markov State Model

The resulting MSM with 200 microstates was obtained using the *k*-means clustering method as described earlier. To validate the Markovian property of the obtained MSM, the implied time scales as a function of the lag time was first plotted; see Figure 4. Clearly, the implied time scale curves level off at lag time around 3 ns, suggesting the convergence of implied time scales with 200 microstates. However, the convergence of implied time scales is not sufficient to show Markovianity [54], because it does not test whether the eigenvectors are also independent of lag time. Hence, C–K test was performed using the *cktest* function in the PyEMMA software [46]. According to the plotted relaxation time scales shown in Figure 4, there were four slow processes separated from others, therefore the C–K test was performed with 5 macrostates and lag time of 3 ns. As shown in Appendix A, the C–K curves estimated from original simulations (black curves) matched quite well with the predicted ones (blue curves) from the constructed MSM, indicating that our constructed MSM is indeed Markovian. Consequently, we lumped all these 200 microstates into 5 macrostates by PCCA+ algorithm based on kinetic similarity to predict the structural dynamics information of TTR monomer over a long time scale.

### 3.3. Structural Ensemble of Key States of Transthyretin Monomer Misfolding

To obtain the detailed structure information for each macrostate obtained from the MSM, 5000 conformations were extracted from each macrostate, and these macrostates are referred to below as S0, S1, S2, S3, and S4. To check the structural change of obtained states relative to the native TTR monomer structure, we plotted the RMSDs distribution of backbone atoms relative to the initial native structure. As shown in Figure 5a, the RMSD of S0 was the smallest with a peak at around 3 Å, while S4 had the largest RMSD relative to the native structure, indicating that S4 is the macrostate with the largest structural change. As for S0, the backbone of the TTR monomer structure was retained quite well. Figure 5b shows a superposition of the representative structure of S0 to the native structure of TTR monomer to validate the well-maintained conformations. As expected, the representative structure of S0 (shown in red) strongly resembled the native one (shown in gray). The position change of conformations in S0, relative to the native structure, is mainly located in the flexible loop regions.

Further, to intuitively examine the structural details of each state, representative structures of each MSM macrostate and the corresponding equilibrium probability are shown in Figure 6. As displayed, except for S0, conformations of all other states were changed more or less, especially for macrostate S4, which is consistent with the above RMSDs distribution as shown in Figure 5. Secondary structure analysis of the β-sheet probability of each residue revealed that D-strand was destroyed in all states. This instability of the D-strand makes it plausible as an initial site for TTR monomer unfolding as reported by Ortore et al. [17]. The other β-strands of S0 were maintained quite well. For S2 and S3, although the RMSDs relative to the native TTR monomer are larger than those for S0 (Figure 5), the β-sheet structures remained quite well, indicating the native-like β-sheet property of these two macrostates. In comparison, β-strands in S1 and S4 were disrupted significantly and partially unfolded into disordered random coil and turn structures as shown in Appendix A, especially for the DAGH sheet, consistent with the significant mobility of DAGH sheet found in the MD simulation study of Jitendra et al. [18]. 

Interestingly, the β-sheet structure of most residues in the CBEF sheet remained even in S1 and S4. However, as shown in Figure 6, there was a tendency for the F-strand to move away from the E-strand and reform a β-sheet structure with the H-strand (see red cycle in the representative structure of S4). These findings are in line with the H/D exchange rates of native TTR tetramer which show a large mobility of strands at edge regions [55], including F-strand and H-strand. Rodrigues et al. also observed the large structural dynamics of F-strand and H-strand by MD simulations. These appear to be the most sensitive regions to thermal unfolding conditions [20]. More importantly, Saelices et al. have previously proposed that F- and H-strands are the aggregation-driving segments of TTR and residue replacements on the F- and H-strands can hinder TTR aggregation [56]. Therefore, we speculate that this newly rearranged structure of F-strand with H-strand may an important conformational state for TTR monomer misfolding.

It is also worth noting that S0, which most strongly resembles the native structure, has the lowest equilibrium probability (about 2.40%), suggesting the instability for this near native state. By contrast, S4, with the largest RMSDs compared to native structure, has the largest population (about 85.28%), which means that this partially unfolded state is extremely stable and may serve as an aggregation-prone state for further TTR aggregation, consistent with the findings of Conti et al. [57]. Therefore, taking the above-obtained results together, we then treated S0 as the native state, while treating S4 with partially unfolded conformations and the largest equilibrium probability as the potential misfolded state to further explore the potential misfolding mechanism of TTR monomer.

### 3.4. Insights into the Misfolding Mechanisms of TTR Monomer

To evaluate the conformational change of the near native state S0 and the potential misfolding state S4, dynamical cross-correlation map (DCCM) was employed (Figure 7). DCCM is useful for finding important structural insights into the function of biomolecules. It appears from Figure 7 that, there was an increase of positive correlations between residues in S4, especially for the region from residue 10 to 76, suggesting a reduced conformational flexibility and tighter packing of residues for conformations in S4. We also calculated the radius of gyration (Rg) of conformations in these two states to confirm the above observation. Consistently, the Rg of conformations in S4 had relatively smaller Rg distribution compared to that of S0 (Appendix A), which indicates the more compact structures in S4. This result is also in line with the previously proposed idea that amyloid formation by TTR is triggered by tetramer dissociation to a compact non-native monomer [10]. Moreover, these compact conformations also explain the extreme stability of S4.

Despite the overall increased positive correlations, a negative correlation from residue 66 to 100 containing strands E and F was observed for S4; negative values are represented with blue color in the DCCM map. This suggests decreased coupling of the E-strand and F-strand, consistent with the outer movement of F-strand from E-strand observed in Figure 6. Correspondingly, increased positive correlations found between residues 90–102 and residues 103–120 (marked by the red ovals in the DCCM map) are in accord with a newly formed β-sheet structure between the F-strand and H-strand. Consequently, compared to the near native state S0, the misfolded state S4 is more compact with obvious structural change.

We also applied transition path theory (TPT) on the constructed MSM to gain a mesoscopic view of the potential misfolding pathway for the TTR monomer into the partially unfolded misfolded state. TPT is a theoretical method to predict folding/unfolding pathways with reactive flux between folded and unfolded states [58]. We used S0 which strongly resembles the native structure, to represent the initial native state of TTR monomer, while S4, the most populated state with partial unfolded conformations, was used to model the misfolded state of the TTR monomer. To distinguish the initial state and final state from other macrostates in the transition pathway, we assigned S0 as SA, and S4 as SB, while, macrostates S1, S2, and S3 remained unchanged. The transition pathway analysis was carried out using the coarse-gained TPT method [59]. A picture describing the relative position of each macrostate in the free energy surface is displayed in Figure 8. From the free energy surface, initial state SA (S0) is located in the lower left corner, while the final state SB (S4), with the lowest energy, is located in the right. In addition, from the free energy surface, it is clear that the energy barrier of interstates was relatively high. This is consistent with the Markovian property of a realistic MSM that interstate transition is slow due to a high energy barrier [43,60].

To visually describe the misfolding events of the TTR monomer, Figure 9, the transition pathways connecting initial native state SA and final misfolded state SB are used to depict the conformational transition. It is clear that there are three pathways for the conformational transition from SA to SB: SA→S1→SB, SA→S3→S2→SB, and SA→S3→S1→SB. Just like the structural characterization we shown in Figure 6, conformations in S2 and S3 maintain their β-sheet structure quite well, while conformations in S1 are partially unfolded with part of DAGH sheet converted into disordered coil and turn structures. Thus, the three pathways can be classified into two categories: (1) conformational transition via native-like β-sheet intermediates (SA→S3→S2→SB) or (2) conformational transition via partially unfolded intermediates (SA→S1→SB, SA→S3→S1→SB). These two categories for the misfolding pathway have been deduced by Anvesh et al. from experimental solution and solid-state NMR measurements [14]. Pathway percentage calculations reveal that it is more likely for the TTR monomer to misfold via partially unfolded intermediates with a percentage of 83.96% as shown in Figure 9. Based on the results of structural characterization and transition pathway network, we speculate that the conformation transition pathway of TTR monomer misfolding is initiated at D-strand, since this strand is quite unstable in all obtained macrostates. Subsequently, other strands in DAGH sheet are partially unfolded to disordered coil or turn structure, and the F-strand moves away from E-strand. Finally, the F-strand forms a new β-sheet structure with H-strand through appropriate conformational rearrangement. 

## 4. Conclusions

We explored the potential misfolding mechanism of the TTR monomer by combining microsecond hybrid-resolution MD simulations and MSM analysis. By comparing PACE simulation and all-atom simulation, we found that the PACE model can provide accurate data and both models revealed that all the loops as well as D-strand in DAGH sheet are quite flexible. From the constructed MSM based on PACE runs, we identified a potential misfolded state of TTR monomer with very large equilibrium probability. By structural analysis, this misfolded state was found to be partially unfolded at the DAGH sheet and was significantly converted into disordered coil and turn structure. In addition, we found that the F-strand could move away from E-strand to form a new β-strand with H-strand in the DAGH sheet, suggesting an important role for both F- and H-strands in TTR misfolding. Further transition pathway analysis indicated that there are two main pathways for TTR monomer to misfold: via native-like β-sheet intermediates or via partially unfolded intermediates. The later pathway is the dominant one for TTR monomer misfolding. Our results have theoretically elucidated the misfolding mechanism of TTR monomer and described the potential misfolding pathway at atomic resolution. This should provide valuable theoretical insights for understanding the TTR aggregation and the pathogenesis of ATTR amyloidosis. 

## Figures and Tables

**Figure 1 biomolecules-09-00889-f001:**
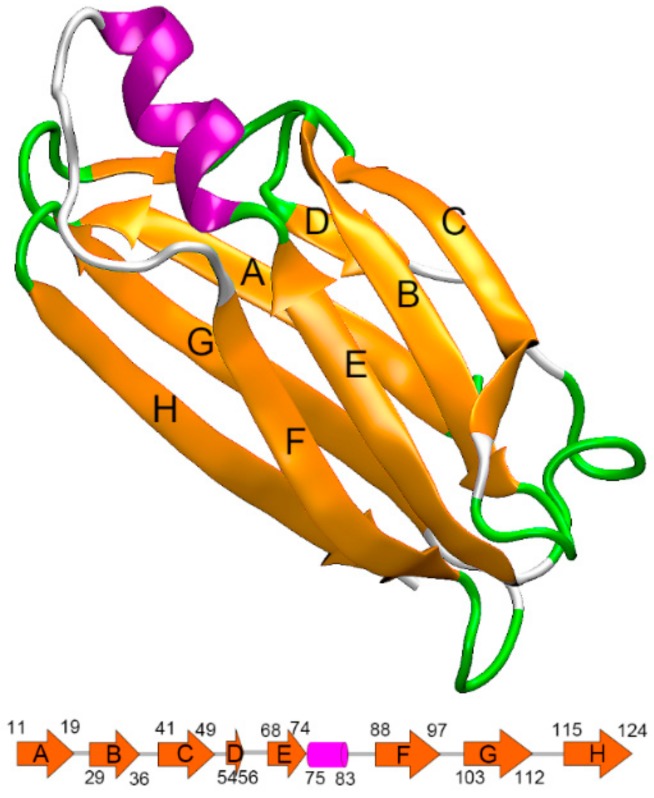
The native structure of transthyretin (TTR) monomer (PDB ID: 4PVM) with labels on all β-strands, constituting two four-β-stranded anti-parallel sheets called the DAGH and CBEF. The cartoon at the bottom shows corresponding residue numbers for each strand.

**Figure 2 biomolecules-09-00889-f002:**
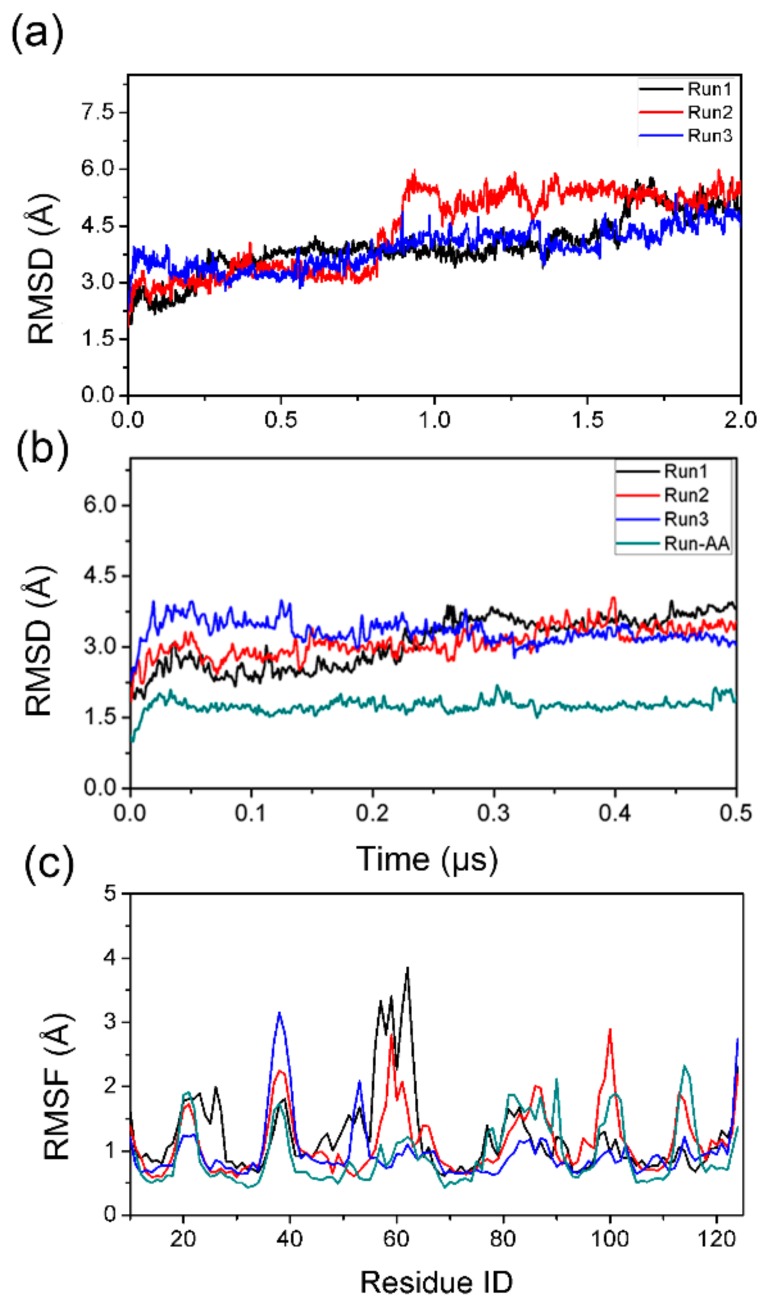
Structural features monitoring of TTR monomer calculated from coarse-grained environment (PACE) (Run1, Run2, and Run3) and all-atom (Run-AA) simulations. (**a**,**b**) The root-mean-square deviations (RMSDs) of Cα atoms relative to the initial TTR monomer structure for three PACE runs and 500 ns all-atom run, (**c**) the RMSFs of Cα atoms as a function of residue number.

**Figure 3 biomolecules-09-00889-f003:**
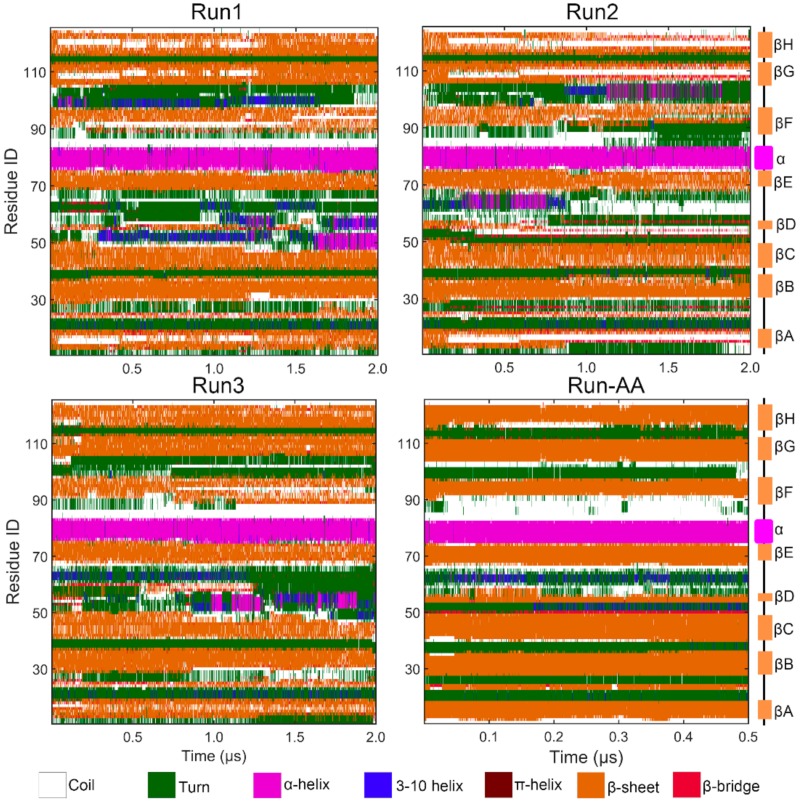
Graphical representation of secondary structure analysis for parallel PACE runs and all-atom run.

**Figure 4 biomolecules-09-00889-f004:**
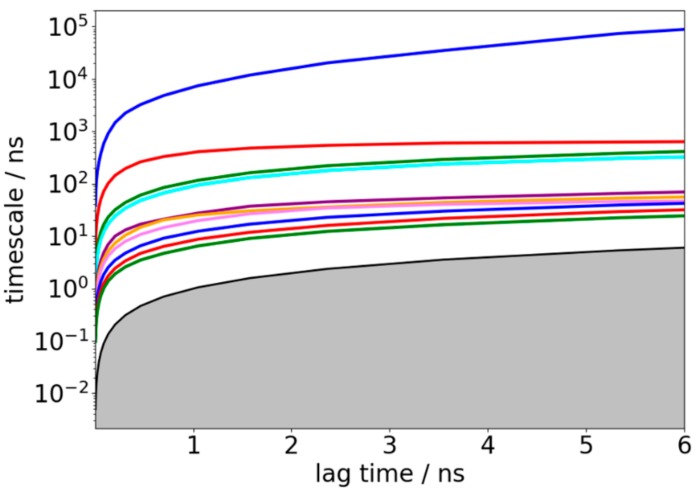
Implied relaxation time scales of the constructed Markov state model (MSM) as a function of lag time.

**Figure 5 biomolecules-09-00889-f005:**
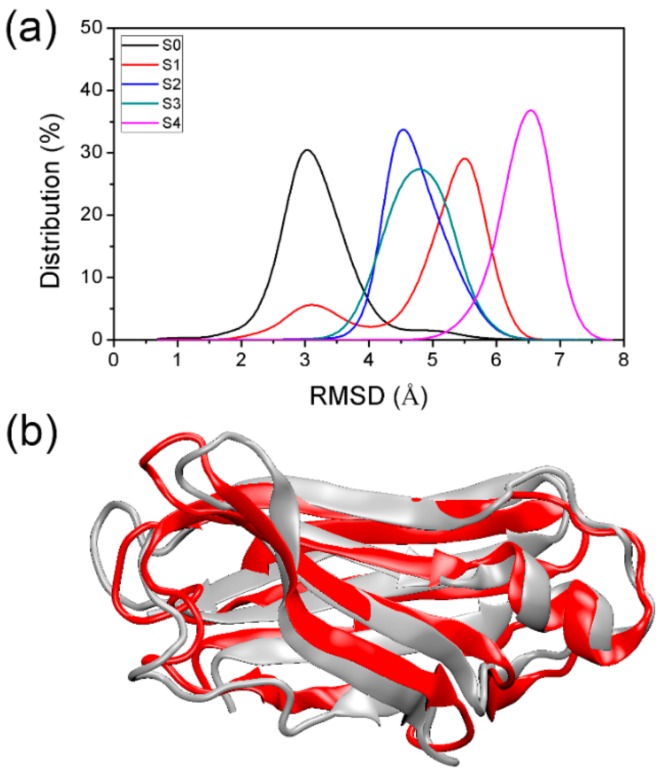
RMSD distribution of backbone atoms of conformations in obtained macrostates relative to the native TTR monomer structure (**a**,**b**) structural superposition of native TTR monomer structure (gray) and representative structure (red) of S0.

**Figure 6 biomolecules-09-00889-f006:**
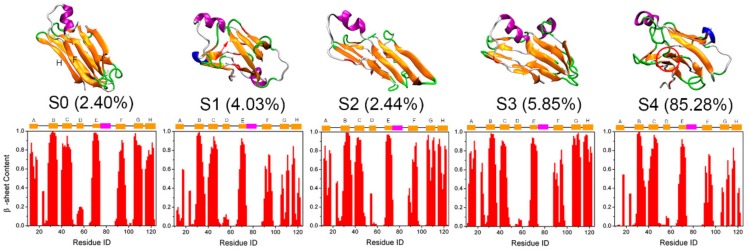
Representative structure of each macrostate obtained from MSM and their corresponding equilibrium probability. The probability for each residue to adopt β-sheet structure within the corresponding macrostates is also displayed below.

**Figure 7 biomolecules-09-00889-f007:**
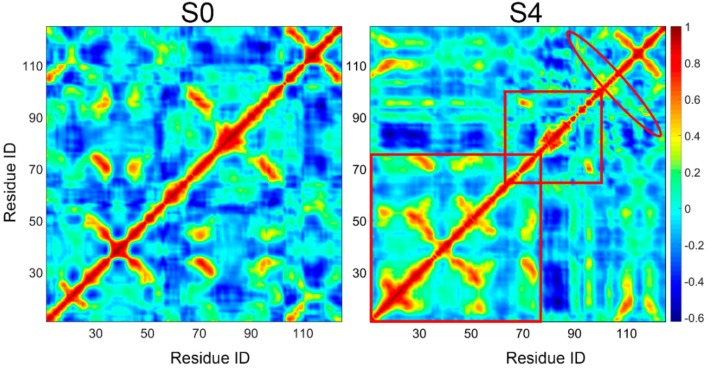
Dynamical cross-correlation matrix analysis for near native state S0 and potential misfolded state S4. Domains with obvious motion difference in two states were labeled with red box and oval.

**Figure 8 biomolecules-09-00889-f008:**
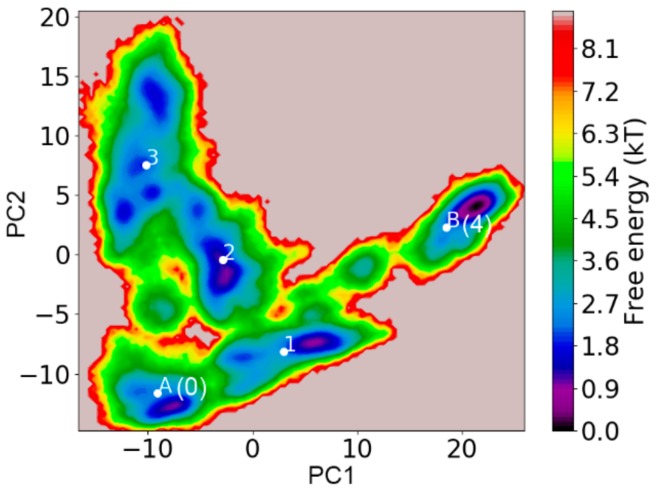
Free energy surface of TTR monomer with average location of corresponding macrostates labeled. A (S0) and B (S4) represent the initial and final states in the transition pathways, respectively. 1, 2, and 3 represent macrostates S1, S2, and S3.

**Figure 9 biomolecules-09-00889-f009:**
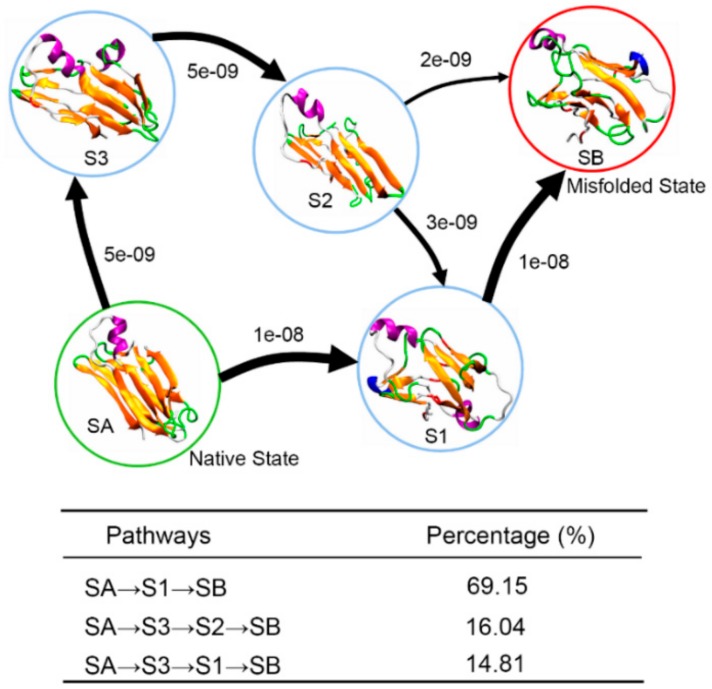
Structural transition pathways of the TTR monomer from the near native state SA (S0) to the potential misfolded state SB (S4). The arrows represent the coarse-gained fluxes with arrow size proportional to the main flux. Percentage of corresponding pathways is shown in the table.

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
