# Peer review of "Exploration of the Misfolding Mechanism of Transthyretin Monomer: Insights from Hybrid-Resolution Simulations and Markov State Model Analysis"

_biomolecules, 2019, doi:10.3390/biom9120889_

Round 1

Reviewer 1 Report

The authors investigated the misfolding mechanism of transthyretin (TTR), using microsecond hybrid-resolution molecular dynamics stimulation and Markov state model analysis. The results elucidated the mechanism of TTR monomer misfolding, describing the potential misfolding pathway at atomic resolution.

This is an important study focusing on the mechanisms of TTR misfolding and aggregation, providing important insights into the pathophysiology of amyloidogenic transthyretin (ATTR) amyloidosis. It is timely because new therapeutic options for this disease, such as TTR stabilizers, small interfering RNA, and antisense oligonucleotide, now appear one after another. The manuscript is well written.

Although I do not have any critical comments, minor issues and suggestions to strengthen this manuscript are raised as follows: 

The authors use “TTR amyloidosis” and “ATTR” for referring “ATTR amyloidosis” in the abstract and main text, respectively. According to a recent nomenclature recommendation, “TTR” and “ATTR” are proteins, not disease (Amyloid 2018; 25: 215-219). Therefore, “ATTR amyloidosis” should be used for disease name.

At the first sentence of the introduction section, the authors mentioned “numerous” ATTR amyloidosis, including senile systemic amyloidosis, familial amyloidotic polyneuropathy, and familial amyloidotic cardiomyopathy. According to a recent definition, ATTR amyloidosis is consisted of wild-type ATTR (ATTRwt) amyloidosis and variant ATTR (ATTRv) amyloidosis. ATTRwt amyloidosis has been traditionally named senile systemic amyloidosis, while ATTRv amyloidosis has been called familial amyloid polyneuropathy or familial amyloid cardiomyopathy (Biomedicines 2019; 7: E11). I would recommend mentioning this issue at this sentence, citing this article, rather than references 1 and 2 in the present version.

Recent studies revealed a difference in the morphology of amyloid fibrils between early- and late-onset ATTR amyloidosis patients (Neurology 2016; 87: 2220-2229; J Neurol Sci 2018; 394: 99-106). Clinical and scientific interests of this manuscript will increase if the authors could discuss the relationship between TTR misfolding and amyloid fibril morphology.

Reviewer 2 Report

The paper focuses on the elucidation of the unfolding mechanism of transthyretin monomer using Hybrid-resolution simulations and Markov State Model Analysis. The paper is generally well written, but a few minor spell check and correction of some sentences is required.

Line 75 "Corazza et al. proposed" although the authors used native mass spec, MD finding data to confirm their NMR data in this paper, I suggest that the original reference is also added: 

Fig 2a. The inset is rather small making the legend quite difficult to read, thus making the interpretation of colour-coded fig 2b rather difficult.

In recent years there has been several papers coming from different groups (.........) suggesting a crucial role played by proteolysis in priming TTR aggregation by destabilisation of the tetrameric native state and release of truncated species together with the full length promoter. Have the authors considered the possibility of investigating the structural behaviour of the truncated protomer before it gets released? i.e. when it is still retained within the tetrameric assembly.

Which pH was set for the simulation? According to the generally accepted mechanism tetramer dissociation is induced by exposure to low pH, although it is not clear where such conditions could be find in vivo. Was the simulation set to physiological or acidic pH?

Fig 8. The free energy surface is often cited in the main text, but it is not clear from the picture what the free energy of the different states actually is. for example is S1 similar to SA in terms of free energy of are they different? Please clarify this point either in the text of the figure legend.
